# Synergetic Adsorption–Photocatalytic Activated Fenton System via Iron-Doped g-C₃N₄/GO Hybrid for Complex Wastewater

**Huihui Mao [1,*], Lu Wang [1], Qing Zhang [1], Feike Chen [1], Yizhou Song [1], Haoguan Gui [1], Aijun Cui [1,2] and Chao Yao [1,*]**

[1]   Jiangsu Key Laboratory of Advanced Catalytic Materials and Technology, School of Petrochemical Engineering, Changzhou University, Changzhou 213164, China

[2]   Analysis and Testing Center, NERC Biomass of Changzhou University, Changzhou 213164, China

*   Correspondence: maohuihui_beijing@126.com (H.M.); yaochao@cczu.edu.cn (C.Y.)

**Abstract:** A synergetic adsorption–photocatalytic-activated Fenton system using an iron-doped g-C₃N₄/GO (GO/Fe-GCN) hybrid with highly efficient performance was established. The highly dispersed iron species with a $Fe^{2+}/Fe^{3+}$ ratio (1.67) and mesopores (3.7 nm) with a relative higher specific area and pore volume benefited the reaction efficiency and the contact of organic pollutants with the active sites. In the dynamic adsorption–photo-coordinated Fenton system, the maximum removal rate of GO/Fe-GCN reached 96.5% and equilibrium was 83.6% for Rhodamine B. The GO component not only enhanced the adsorption but also provided a higher efficiency of photo-generated carrier separation and transport. The hybrid structure of GO/Fe-GCN and the high efficiency of circulation of Fe(III)/Fe(II) played an essential role in the synergy of the adsorption–enrichment and the photo-coordinated Fenton reaction. GO/Fe-GCN can also be used to treat complex waste-water containing metallic ions, metal complexes, and organic pollutants, which could allow potential applications in the treatment of water pollution.

**Keywords:** synergetic adsorption-photocatalysis; iron-doped g-C₃N₄/GO hybrid; photocatalytic activated Fenton; dynamic adsorption-photo-coordinated Fenton





## 1. Introduction

Graphite-phase carbon nitride (g-C₃N₄) is a high quality versatile two-dimensional material found among the visible light photo-catalysts [1]. However, the photo-catalytic efficiency of pure g-C₃N₄ is limited, due to disadvantages such as its small specific surface area [2,3], limited light response [4], and low photo-electron-hole separation efficiency [5–7]. For photo-catalysis processes, most reactions occur at the interface of the photo-catalysts, and the nano-scale structure, porosity, and interfacial performance of photocatalysts have a great influence on the photocatalytic activity [8–10]. The design of the nanostructure should, not only increase the specific surface area of g-C₃N₄ and expose more effective areas and active sites for the photocatalytic reaction procedure, but also improve the light capture ability of the material. Previously reported designs for g-C₃N₄ nanostructures include quantum dots, one-dimensional nanofibers/nanowires/nanoribbons [11–13], two-dimensional ultrathin nanosheet/porous nanosheets [14,15], and three-dimensional nanoflower/hollow nanospheres [16–20].

For water pollution treatment, adsorption is a widely applied, low-cost, and high-efficiency method. Thus, the adsorption method is adopted at large-scale for environmental protection. However, the adsorption method has several disadvantages that limit the usage of certain adsorbents. The adsorbents need to be recovered after a certain time, because the pollutants that become concentrated in them will not degrade [21]. In addition, the adsorption capacities of adsorbents are limited. In order to solve these problems, the synergistic strategy of adsorption–photocatalysis was developed in the field of pollutant

water treatment [22–25]. The critical factor of this synergistic strategy is to assemble suitable materials, with both high adsorption capacities and excellent photo-catalytic activities. The pollutants are adsorbed quickly in special structures and then degraded in situ through photo-catalysis. This interesting field has attracted much attention from researchers, and various methods, including hydrogel [26], layered structure [27], 3D structure [28], porous structure [29], and heterojunction construction, have been developed [30,31].

Fenton and photo-activated Fenton procedures are widely used and efficient wastewater treatment technologies [32]. The recycling of $Fe^{2+}/Fe^{3+}$ in the presence of hydrogen peroxide generates numerous hydroxide radicals ($\bullet OH$), resulting in the rapid degradation of pollutants in an aqueous phase [33]. Nevertheless, as an important advanced oxidation process (AOP), the traditional Fenton process still suffers from drawbacks, such as mud containing $Fe^{3+}$ salts and a narrow pH range. Therefore, heterogeneous iron-based Fenton catalysts, with the advantages of a high efficiency, low cost, wide pH value range, and reusability, have become the focus of wastewater treatment [34,35].

Coupling adsorption and an advanced oxidation process (AOP), two widely utilized water treatment technologies, is an attractive method for complex wastewater treatment. The adsorption process removes a wide range of pollutants from the aqueous phase. However, the pollutants are only transferred from one phase to another, rather than being transformed to harmless end products [36]. With the AOP, the adsorbed pollutants can be degraded or even effectively mineralized using oxidative radicals such as hydroxyl radicals ($\bullet OH$) and sulfate radical ($SO_4\bullet^-$) that are generated in situ in an advanced oxidation process (AOP). Combining the adsorption with AOP leads to a significant enhancement in treatment efficiency, especially when the catalytic oxidation is carried out on the surface of the adsorbent where the pollutants are concentrated [36–41].

The objective of this research was to synthesize and explore the mechanism of an iron-doped $g$-$C_3N_4$/GO hybrid composite for a synergetic adsorption–photocatalytic activated Fenton system with highly efficient performance. Inspired by $Fe$-$g$-$C_3N_4$/graphitized mesoporous carbon [42], mesoporous magnetite/carboxylate-rich carbon composites [43], Cu-Zn-Fe-LDH [44], and nano-FeO(OH)/reduced graphene oxide aerogel [45], the combination of iron-doped $g$-$C_3N_4$ with GO resulted in an efficient photo-activated Fenton catalyst that functioned in a wide pH range. $Fe^{2+}$ and $Fe^{3+}$ were trapped by N-rich $g$-$C_3N_4$, which formed highly-dispersed active sites. Meanwhile, GO not only provided a high adsorption capacity for the $g$-$C_3N_4$/GO hybrid composite, but also promoted the electron transfer via the strong electronic coupling of the $sp^2$ bonding structure [46].

## 2. Results and Discussion

### 2.1. Structural Characterization of Samples

GO was synthesized using the Hummers method [47,48]. The Fe-GCN was assembled from $Fe(NO)_3 \cdot 9H_2O$ and diccyanide solution in a water bath at 80 °C (As shown in Figure 1a). Samples of Fe-GCN and GO/Fe-GCN are shown in Figure 1b. The GO/Fe-GCN presents a typical sheets-like structure in the SEM and TEM images (Figure 1c,d); the STEM image (Figure 1f) and overlapping of iron, nitrogen, and carbon EDX maps demonstrate the uniform distribution of iron element with nitrogen and carbon. The absence of obvious Fe clustering reveals that the Fe species were uniformly dispersed in the GO/Fe-GCN, rather than simply mixing at nanoscale. The SEM image of GO (Figure S1a) reveals a single gossamer layer or several layers of ultrathin nano-film. Fe-GCN (Figure S1b) presents a bit thick irregular thin nano flake, showing conjugate aromatic system orderly accumulation. For GO/Fe-GCN (Figure 1e), which is consisted by smooth flake of Fe-GCN and corrugated ultrathinc GO film. In the unique combine structure, nanoscale channels and porous structure are easily formed, which would increase the specific surface area and electron transport efficiency [49].

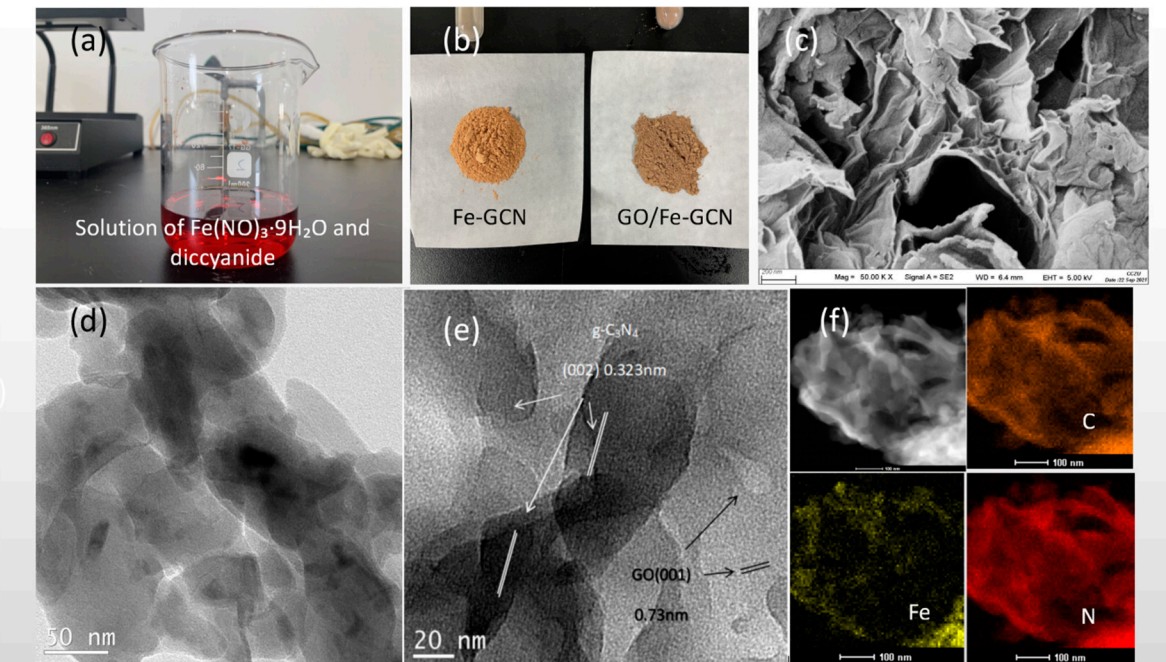

**Figure 1.** Solution of Fe(NO₃)₃·9H₂O and diccyanide (**a**), image of Fe-GCN and GO/Fe-GCN (**b**), SEM image of GO/Fe-GCN (**c**), TEM image of GO/Fe-GCN (**d**), HRTEM of GO/Fe-GCN (**e**), and STEM image with overlapping of Fe, C, and N elements (**f**).

The XRD patterns for Fe-GCN with different proportions of iron are indicated in Figure S2, located at 13.7° and 27.3°, respectively, corresponding to the characteristic peaks of the triazine ring unit (100) crystal face and typical interlayer stack (002) crystal plane of conjugate aromatic system, and corresponding to the XRD diffraction at 2θ of 13.7° and 27.3°, respectively. With the increase of proportion of iron, the diffraction peaks of the crystal planes of (100) and (002) decreased and widened, indicating that the crystallinity decreased and the layers were tightly packed and strongly bonded, which was due to the influence of high temperature thermal condensation of the complex formed by the interaction between dicydiamide and $Fe^{3+}$ [50]. In addition, no iron oxide diffraction peaks were founded, indicating that iron exists in an amorphous form or Fe-N ligand structure. Figure 2a shows the XRD spectrum of the samples. It can be observed that there is little difference in the overall peak pattern of Fe-GCN after recombination with GO, but there is weak diffraction peaks of the (001) and (100) crystal plane, unique to GO and GCN, at about 9.7° and 13.7°, respectively, indicating that GO and Fe-GCN were combined successfully.

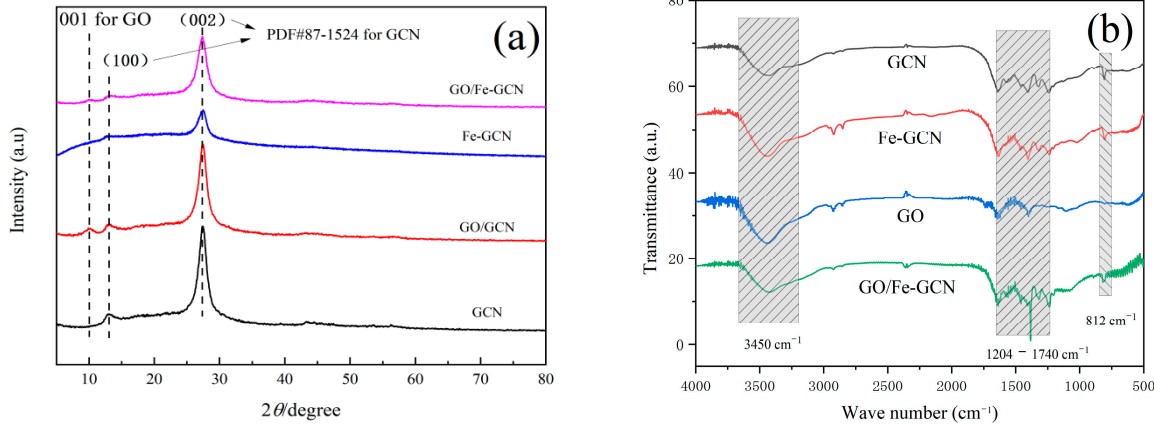

**Figure 2.** XRD spectrum of the samples (**a**), and FT-IR patterns for samples (**b**).

As shown in Figure 2b, the peak around 1204–1740 cm$^{-1}$ for g-C$_3$N$_4$ corresponds to the C-N heterocyclic. The peak at 810 cm$^{-1}$ is attributed to the stretching vibration of the triazine unit, while the large and wide peak at 3450 cm$^{-1}$ is the stretching vibration and bending vibration of −OH. With an increase of iron doping, the peak value at 810 cm$^{-1}$ became weaker, because the Fe-N ligand formed by the doping of iron and the condensation of dicyandiamide reduced the stretching vibration of the triazine ring or heptamine ring. There was a new weak peak at 580 cm$^{-1}$, which may have been from the Fe-O bond, but this peak was obvious only when the doping amount reached 0.5. These results indicated that Fe doping did not affect the main structure of g-C$_3$N$_4$. For GO/Fe-GCN, in the range of 1638 cm$^{-1}$–1233 cm$^{-1}$ the spectra show stronger peaks than Fe-GCN. This enhancement is attribute to GO's oxygen-containing groups (C-O, O=C-O, O=C) [51].

Figure 3 shows XPS patterns for Fe-GCN and GO/Fe-GCN for each element. In the C 1s spectrum for Fe-GCN (Figure 3a), two main characteristic peaks for C-C/C=C sp$^2$ (284.90 eV) and N=C-N (287.96 eV) were observed, which is the typical g-C$_3$N$_4$ structure. In contrast, GO/Fe-GCN (Figure 3b) shows four peaks of C-C/C=C sp$^2$, N=C-N, C-O, and C=O bonds at about 284.73, 287.73, 287.03, and 288.4 eV, respectively. Here, the binding energy of the C-C/C = C sp$^2$ bond is the characteristic peak of GO and structural defects at the edge of g-C$_3$N$_4$. The binding energies of C-O and C=O are from the GO surface. This indicates that N=C-N bonding and C-O carbon atoms were the main species in the GO/Fe-GCN composite. The spectral peak type of N 1s in Figure 3a,b is basically unchanged, the peaks at 400.45 eV, 399.17 eV, and 398.32 eV correspond to the quaternary nitrogen bond of three carbon atoms in the aromatic ring, the tertiary nitrogen bonded by N-(C)$_3$ or H-N(C)$_2$, and sp$^2$ hybrid aromatic N bonded by C=N-C, respectively. Compared with pure carbon nitride, the peak is slightly shifted, which may have been related to the strong interaction between Fe and g-C$_3$N$_4$, thereby accelerating the electron transfer [52]. The BE value of coordination between N and Fe (N-Fe) was 398.8 ± 0.5 eV, in which pyridine N and C/N-Fe were the dominant species. Pyridine N is attributed to the nitrogen–carbon carrier interaction and can be used as a link between the carbon nitride polymer and carbon sheet. The 532.34 eV of O 1s corresponds to the C-O bond for both GO/Fe-GCN and Fe-GCN. The Fe 2p spectrum at 710.1 eV and 719.35 eV correspond to Fe$^{2+}$ 2p$_{3/2}$ and Fe$^{2+}$ 2p$_{1/2}$, and 711.6 eV and 721.91 eV correspond to Fe$^{3+}$ 2p$_{3/2}$ and Fe$^{3+}$ 2p$_{1/2}$, respectively. This shows that the iron species in the materials were in the form of ferrous iron and ferric iron with a ratio of 1.67 (Fe$^{2+}$/Fe$^{3+}$) [53].

As shown in Figure 4a, the optical properties of the samples were characterized using UV-vis DRS, and the absorbance was observed for all samples in the range of 300 to 800 nm. GCN is clearly shown at about 460 nm on the absorption edge. With the increase of iron content, Fe-GCN showed a significant red shift. This phenomenon indicates that the binding structure of the Fe-N bond can improve the utilization of solar energy and enhance the catalytic activity under visible light conditions. As the absorption intensity increased, the absorption edge also increased with the increase of Fe content, which leads to more photogenerated electron–hole pairs in the visible spectrum and higher yields [54]. The band-gaps of the GCN and iron-doped GCN samples were in the range 2.26~2.60 eV (Figures S3 and 4b). More interestingly, compared with the other samples, the band-gap of Fe-GCN-0.15 was dramatically shortened. This reduced band-gap after iron doping can be attributed to the interaction between g-C$_3$N$_4$ and the iron atom. Semiconductors with higher photocatalytic activities usually have a higher efficiency of photo-generated carrier separation and transport. Both Fe-GCN and GO/Fe-GCN belong to typical n-type semiconductors, with positive slope values on the Mott–Schottky curve (as shown in Figure S4a,b). In general, for n-type semiconductors, the flat-band potential is approximately at the conduction band potential (Table S1). As shown in Figure S4c, GO/Fe-GCN displayed the fastest electron transfer rate, which is the reason for its high efficiency of photo-generated carrier separation [55]. Compared with Fe-GCN, the photoluminescence intensity of GO/Fe-GCN was relatively weaker (as shown in Figure S4d). The above results indicate that the recombination of photo-generated carriers was efficiently pre-

vented; however, there was no evidence for heterojunctions being formed between Fe-GCN and GO.

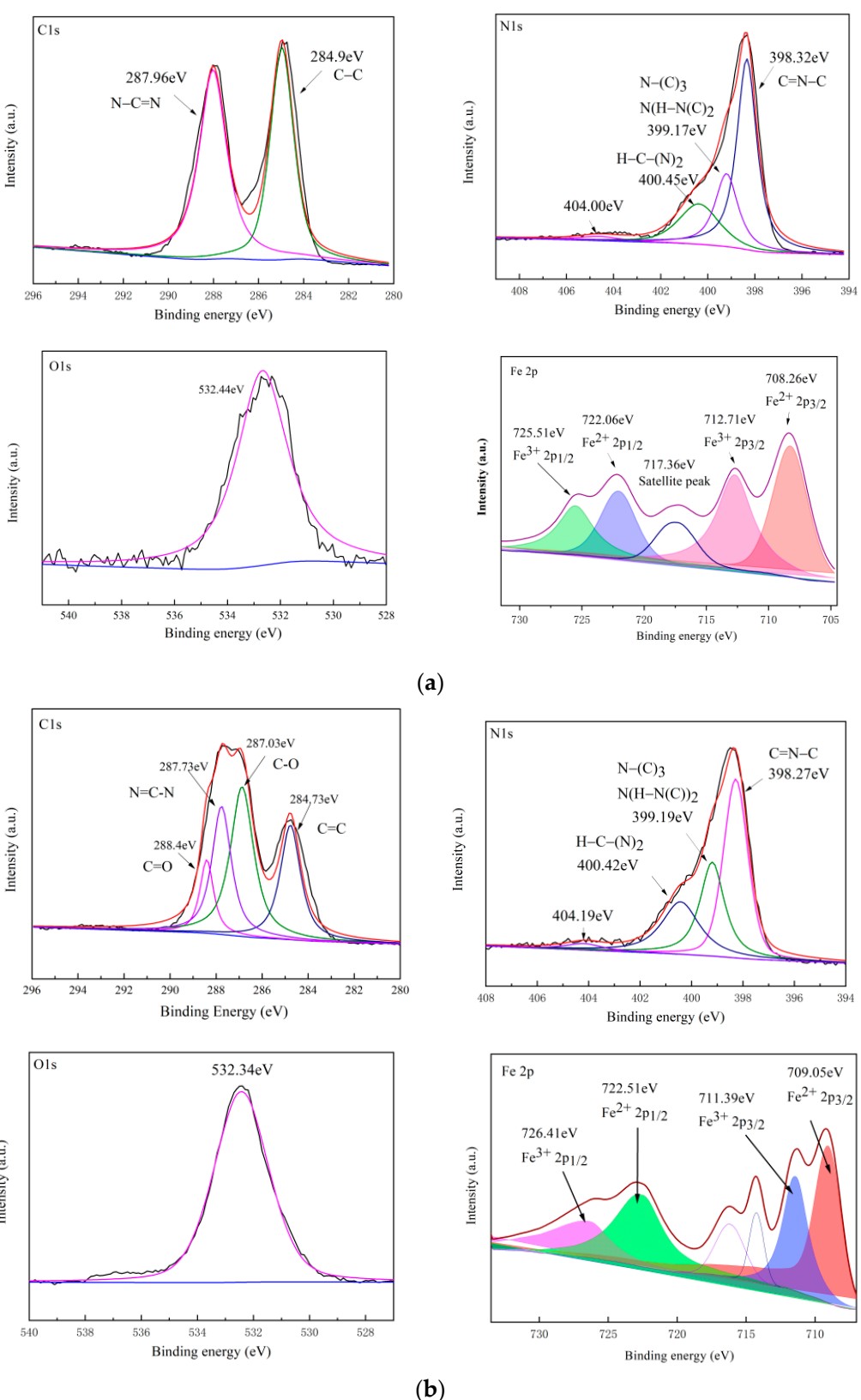

**Figure 3.** XPS patterns for Fe-GCN (**a**) and GO/Fe-GCN (**b**): high-resolution C 1s, N 1s, O 1s, and Fe 2p spectra.

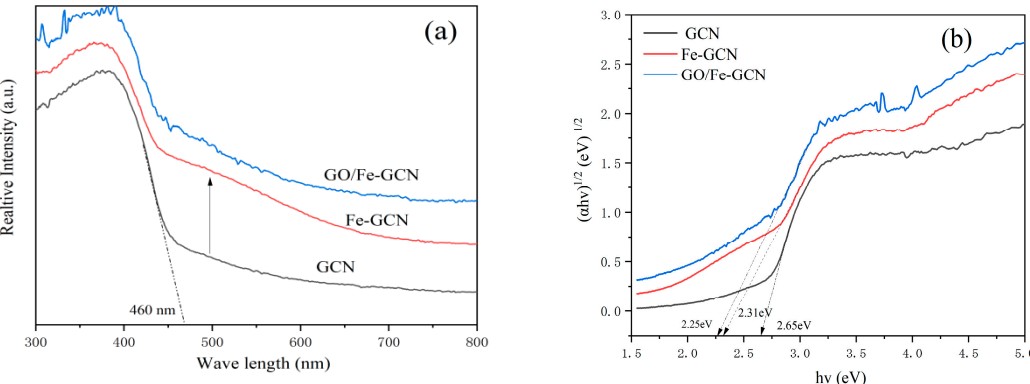

**Figure 4.** UV-vis DRS patterns (**a**) and band-gaps (**b**) of GCN, Fe-GCN, and GO/Fe-GCN.

Figure 5a,b illustrate the $N_2$ absorption–desorption isotherms and BJH pore size distributions of GCN, Fe-GCN, and GO/Fe-GCN, respectively. The $N_2$ absorption–desorption isotherms of GCN, Fe-GCN, and GO/Fe-GCN belonged to the typical IV isotherm or $H_3$ hysteresis loop, indicating the typical mesoporous structure and capillary condensation phenomenon of the catalysts. However, the iron content affected the porous structure of the samples and decreased the specific area and pore volume (As shown in Table S2). The specific area (56.2 $m^2$/g) and pore volume (0.17 $cm^3$/g) of GO/Fe-GCN were larger than in GCN and Fe-GCN, which would benefit the catalytic performance. After combining Fe-GCN and GO, the BJH plot demonstrates the presence of well-established mesopores of 3.7 nm. The pore size of GCN and Fe-GCN was determined in the range of 1~2.4 nm, while that of GO/Fe-GCN was around 3.7 nm. This larger pore size would benefit the contact of organic pollutants with the active sites on the ternary catalyst of GO/Fe-GCN. This shift of the pore size to the mesopore range was attributed to the mesopore defect on GO, which formed during the oxidation process [56].

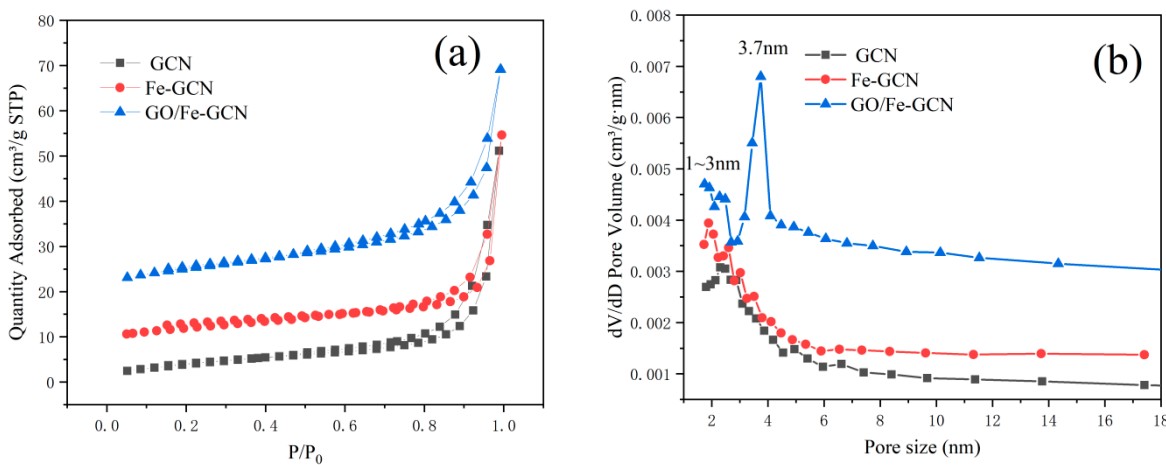

**Figure 5.** $N_2$ adsorption-desorption isotherms (**a**) and BJH pore size distributions (**b**) for GCN, Fe-GCN, and GO/Fe-GCN.

## 2.2. Adsorption-Photocatalytic Properties of Samples

The visible light photocatalytic degradation rate of GCN into Rh B was only 17.5% within 3 h, as shown in Figure 6a, while the photocatalytic degradation rate of Fe-GCN with different iron contents was between 20 and 26.6% (See Figure S5a). Moreover, the adsorption properties of GCN and Fe-GCN samples were limited compared with GO/Fe-GCN (See Figure S5a). The quasi-first-order reaction kinetic rate of Fe-GCN was 1.83 times higher than that of GCN, and the degradation efficiency was slightly improved compared with that of GCN (Figures 6b and S5b). This was because the Fe-N ligand formed by Fe doping reduced the electron hole recombination efficiency and carrier migration. For

GO/Fe-GCN, the degradation efficiency increased to 47.56% within 3 h, corresponding to a 0.0060 min$^{-1}$ first order reaction kinetics rate, which was 5.00 and 2.73 times higher than that of GCN and Fe-GCN. The GO of single-layer material with conductive ability sped up the electron transfer, which is demonstrated in the electrochemical impedance spectroscopy (EIS) image in Figure S4c. Furthermore, it may increase the lifetime of charge carriers, store electrons, and shuttle them to the adsorbed substances on the material surface.

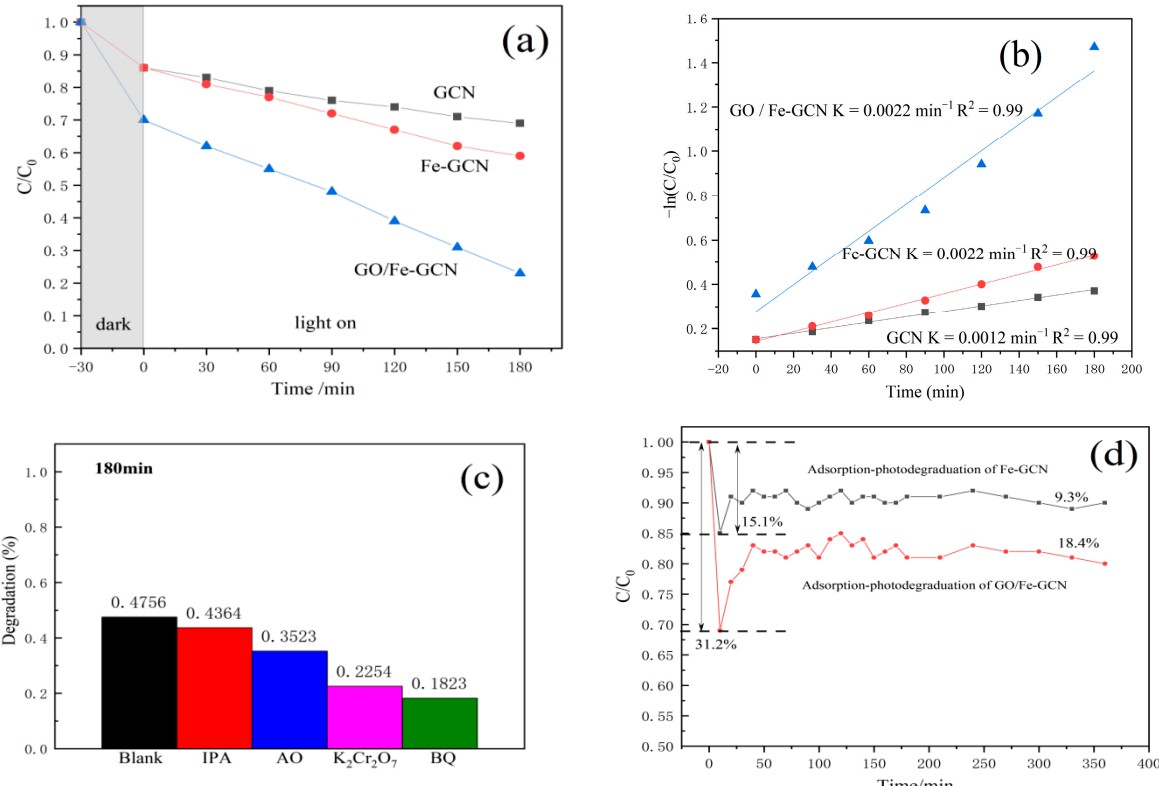

**Figure 6.** The visible light photocatalytic degradation rate of GCN, Fe-GCN, and GO/Fe-GCN for Rh B (**a**). The quasi-first-order reaction kinetic rate of GCN, Fe-GCN, and GO/Fe-GCN towards Rh B (**b**). Addition of active substance catcher in the quenching experiment for GO/Fe-GCN (**c**). The synergy of adsorption–photocatalytic degradation performance of Fe-GCN and GO/Fe-GCN in the dynamic system (**d**).

The main active substances in the photocatalytic system can be revealed by adding active substance catcher in the quenching experiment, so as to study the reaction mechanism. The hydroxyl radicals ($\cdot OH^-$), holes ($h^+$), superoxide radicals ($\cdot O^{2-}$), and electrons ($e^-$) in the system were scavenged by isopropanol (IPA), ammonium oxalate (AO), and potassium p-benzoquinone (BQ) dichromate, respectively. For the photocatalytic system reaction after 3 hours of degradation, as shown in Figure 6c, the effect order of the free radical quenching was $O^{2-} > e^- > h^+ > \cdot OH^-$, which revealed the main active material in the photo-catalytic system for the super oxygen free radicals.

Figure 6d shows the synergy of adsorption–photocatalytic degradation performance of Fe-GCN and GO/Fe-GCN in the dynamic system. It took about 120 min to reach equilibrium in the dynamic system. The maximum removal rate of GO/Fe-GCN reached 31.2%, and the equilibrium removal rate was around 18.4%, both of which were higher than for Fe-GCN. These results indicate the elevated synergistic effect of adsorption and photocatalytic performance with GO/Fe-GCN.

### 2.3. The Synergistic Effect of Adsorption Pre-concentration and Subsequent Photo-Coordinated Fenton Degradation of Samples

The traditional Fenton method utilizes $Fe^{2+}$ and $H_2O_2$ to conduct advanced oxidation processes (AOP) in a homogeneous system. Herein, Fe-GCN and GO/Fe-GCN were applied as heterogeneous photo-coordinated Fenton catalysts. Figure 7a shows that, in the presence of $H_2O_2$, Fe-GCN in visible light had a good degradation performance within 30 min. The quasi-first-order reaction kinetic rate of GCN, Fe-GCN, and GO/Fe-GCN under photo-coordinated Fenton degradation is shown in Figure 7b. The photo-coordinated Fenton degradation efficiencies of GCN, Fe-GCN-0.05, Fe-GCN-0.15, Fe-GCN-0.25, and Fe-GCN-0.5 were 29.44%, 52.95%, 86.71%, 58.71%, and 76.6%, respectively (See Figure S6). With the increase of the Fe doping amount, the degradation efficiency of the photo–Fenton technique was enhanced. However, too high a Fe content caused a decline of the degradation efficiency, and this was attributed to excessive iron nitrate as a precursor not favoring dispersion of Fe in the material through the calcining process. The excess of Fe species covered the active sites of the catalysts. The photo-coordinated Fenton degradation rate of GO/Fe-GCN was enhanced to 98.71% within 20 min, because the excellent conductivity of GO accelerated the electron transport rate and $Fe^{2+}/Fe^{3+}$ circulation efficiency [57].

Under photo-Fenton conditions, different captured agents had an inhibitory effect on the degradation reaction (Figure 7c), with IPA, AO, potassium dichromate, and BQ degraded after 30 min. The degradation rates were 10.34%, 90.78%, 84.45%, and 82.98% in the presence of IPA, AO, potassium dichromate, and BQ, respectively. These results indicate that hydroxyl radicals play a major role in a photo-Fenton system. Photo-excited electrons combined with $H_2O_2$ in the solution, to form hydroxyl radicals. As an excellent conductive material, GO enhanced the transmission of electrons. Therefore, the higher efficiency of photo-generated carrier separation and transport was attributed to the presence of GO, which is consistent with the photo-Fenton experiment data.

Electron paramagnetic resonance (EPR/ESR) can be used to study free radical active species. This characterization was performed by trapping hydroxyl radicals with the free radical trapping reagent DMPO, to form the complex DMPO-$\cdot OH^-$. Under either dark or light conditions, maps are presented with four-times the size of the typical hydroxyl radical peak and a peak ratio of 1:2:2:1, in accordance with the quenching results (Figure 7d). In addition, the peak value under dark conditions was obviously weaker than that under light conditions, confirming that visible light (Xenon lamp) played an excitation role and enhanced the efficiency of the reaction [58].

The synergistic effect of the adsorption pre-concentration and subsequent photo-coordinated Fenton degradation of samples was investigated in a dynamic system (as shown in Figure 7e).

The GO/Fe-GCN showed strong adsorption–enrichment in the dark, and it took about 120 min to reach adsorption saturation. The maximum adsorption removal rate of GO/Fe-GCN reached 31.3%, while that of Fe-GCN was only 7.9%. For the adsorption–photo-coordinated Fenton system, the maximum removal rate of GO/Fe-GCN reached 96.5%, with equilibrium at 83.6%. This result greatly exceeded the only adsorption process and photo-coordinated Fenton degradation of Fe-GCN. This synergistic effect contributed to the adsorbing and concentrating the low concentration pollutant in water, and then the enriched pollutant could be treated with photo-coordinated Fenton-degradation in situ. The XRD pattern after 300 min adsorption photo-coordinated Fenton degradation is shown in Figure 7f), which was basically unchanged, indicating that the structure of the GO/Fe-GCN was not deformed during long-time utilization.

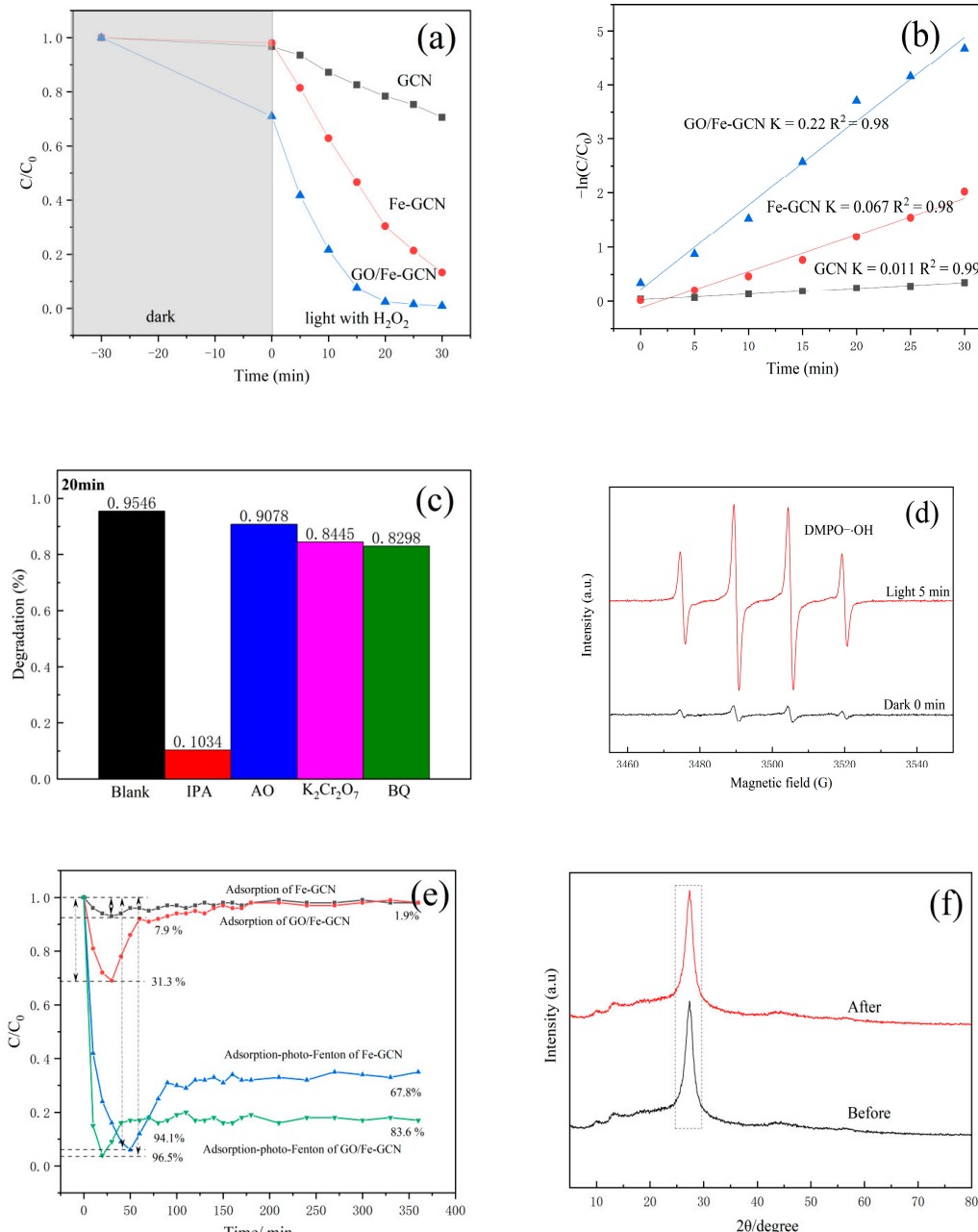

**Figure 7.** The photo-coordinated Fenton degradation efficiencies of GCN, Fe-GCN, and GO/Fe-GCN (**a**). The quasi-first-order reaction kinetic rate of GCN, Fe-GCN, and GO/Fe-GCN (**b**) under photo-coordinated Fenton degradation. The various captured agents with the inhibitory effect on the degradation reaction (**c**) and electron paramagnetic resonance (EPR/ESR) (**d**) for GO/Fe-GCN. The dynamic adsorption and synergy effect of adsorption- photo-coordinated Fenton degradation on the dynamic system of samples (**e**). XRD patterns of GO/Fe-GCN before and after the adsorption–photo-coordinated Fenton degradation process (**f**).

Based on the quenching and EPR experiments and other results above, the photo-Fenton catalytic mechanism of the reaction system was proposed (as shown in Figure 8). First, organic pollutants were efficiently adsorbed and enriched on the surface of the GO/Fe-GCN, which shortened the mass transfer distance for the subsequent Fenton process. The Fe(II) in GO/Fe-GCN generated ·OH and OH$^-$ with $H_2O_2$, and Fe(II) was transmitted into Fe(III) (Equation (1)). The GO/Fe-GCN with a band gap of 2.25 eV was activated under visible light. Visible light caused the separation of photo-generated electrons and holes in

the semiconductor material (Equation (2)). The photo-excited electrons were transferred to the wrapped GO layer, due to its facile charge transporting nature (Equation (3)) [59]. Most of the electrons in the GO layer reduced the Fe(III) into Fe(II) (Equation (4)), and the electron transfer led to the REDOX circulation between Fe(III) and Fe(II). The photo-induced holes could not be transformed to the hydroxyl radicals with $OH^-$, owing to the valence band of GO/Fe-GCN (1.76 eV) being lower than the potential of $OH^-/OH$ (2.29 eV) [60]. However, the holes had a strong ability for oxidative degradation and served as the active species in the degradation reaction. The GO in the composite accelerated the velocity of electron transfer and some electrons could be transferred to GO by $\pi$-$\pi$ conjugation, to promote electron decomposition of $H_2O_2$ to form OH [61]. Furthermore, the structure of GO provided more transfer channels, to facilitate the transformation of the electrons and holes. The above mechanism reveals that the hybrid structure of GO/Fe-GCN and the high efficiency circulation of Fe(III)/Fe(II) play an essential role in the synergy of the adsorption-enrichment and photo-coordinated Fenton reaction.

$$Fe(II) + H_2O_2 \rightarrow Fe(III) + OH^- + \cdot OH^- \tag{1}$$

$$GO/Fe\text{-}GCN + h\nu \rightarrow e^-_{CB} + h\nu^+_{VB} \tag{2}$$

$$e^-_{CB} + GO \rightarrow GO^- \tag{3}$$

$$Fe(III) + GO^- \rightarrow Fe(II) + GO \tag{4}$$

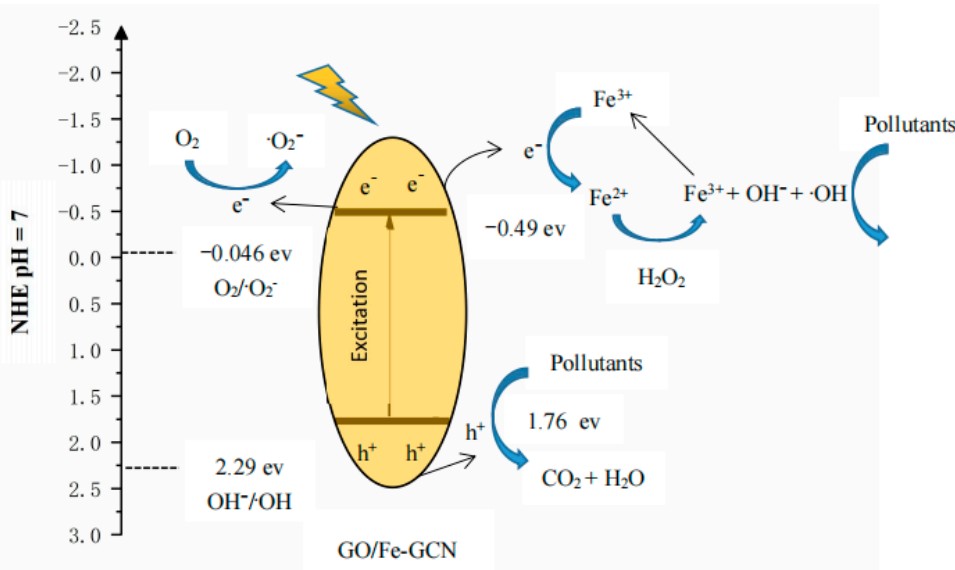

**Figure 8.** Photo-Fenton catalytic mechanism of GO/Fe-GCN in the reaction system.

In the traditional homogeneous Fenton process, hydroxyl radicals are generated in the process of $Fe^{2+}$ oxidation to $Fe^{3+}$, and then reduced to $Fe^{2+}$ via another hydrogen peroxide molecule. However, the single $\cdot OH^-$ generation channel consumes a large amount of $H_2O_2$. Moreover, the $Fe^{2+}$ in the system is hard to recycle, resulting in a large amount of resource waste and iron-containing sludge. GO/Fe-GCN, a heterogeneous catalyst, is easily separated from the liquid phase. The recycling process only requires filtering followed by drying and collection, without any extra treatment. Moreover, the GO/Fe-GCN remains highly active after five cycles (See Figure S7).

### 2.4. The Adsorption Pre-concentration and Subsequent Photo-Coordinated Fenton Process for Complex Wastewater

The synergistic effect of the adsorption pre-concentration and subsequent photo-coordinated Fenton degradation of GO/Fe-GCN is widely and satisfactorily utilized, not only for organic pollutants (as shown in Figure 9a), but also for complex pollutants such

as metal complexing compounds and waste water containing both metals and organics. Metallic ions can be removed by traditional processes in aqueous phase, such as precipitation, adsorption, and ion exchange. However, metallic ions tend to receive an electron pair from some functional groups, which form stable organic complexes. The formed complexes enhance the mobility of metals in the aqueous phase and make them resistant to the traditional methods mentioned above. The removal of $Cu^{2+}$-EDTA is illustrated in Figure 9b, where it can be seen that $Cu^{2+}$-EDTA was hard to remove only with adsorption, suggesting that $Cu^{2+}$-EDTA is stable in aqueous phase. With visible light and $H_2O_2$, the removal efficiency was increased to 94.5% within 30 min, demonstrating the excellent degradation performance of the Cu-complex. Notably, the removal efficiency of $Cu^{2+}$-EDTA and $Cu^{2+}$ was out of step, and that of $Cu^{2+}$ was relatively lower. This was due to the fact that $Cu^{2+}$-EDTA was attacked by $\cdot OH^-$ and the molecular structure of organic EDTA was destroyed. Then the $Cu^{2+}$ was released from the stable complex structure. The free $Cu^{2+}$ was adsorbed by GO/Fe-GCN [62,63]. The simultaneous reduction of Cr(VI) and degradation of Rh B is shown in Figure 9c, it is interesting to note that limited Cr(VI) was reduced in a single system (as shown in Figure S8). With the coexistence of Rh B, the reduction efficiency of Cr(VI) reached 96.7% within 30 min, which is even higher than the degradation efficiency of Rh B. First, some Rh B and Cr(VI) were adsorbed onto the surface of GO/Fe-GCN. Part of the Cr(VI) and $Fe^{3+}$ was reduced by photo-induced electron. The hydroxyl radicals were generated in the process of $Fe^{2+}$ oxidation to $Fe^{3+}$, and $Fe^{2+}$ itself could also reduce Cr(VI) into Cr(III). The organic pollutant was attacked by hydroxyl radicals and photo-generated holes; it was be converted into reductive intermediates such as oxalic acid, which could also reduce Cr(VI) to Cr(III) (as shown in Figure 10). This process resulted in a slightly lower Rh B degradation efficiency in the mixed solution than the reduction rate of Cr(VI) [64,65].

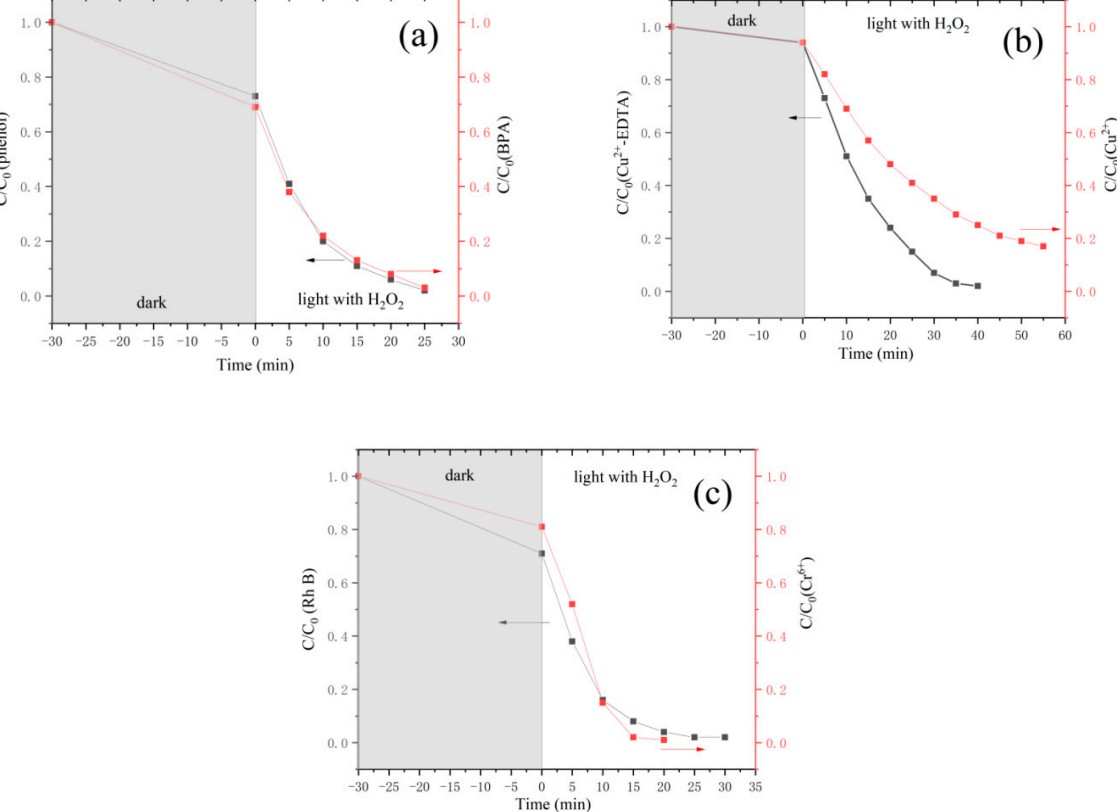

**Figure 9.** The synergistic effect of adsorption pre-concentration and photo-coordinated Fenton degradation of GO/Fe-GCN for phenol and BPA (**a**), Cu-EDTA (**b**), and complex pollutants containing Rh B and $Cr^{6+}$ (**c**).

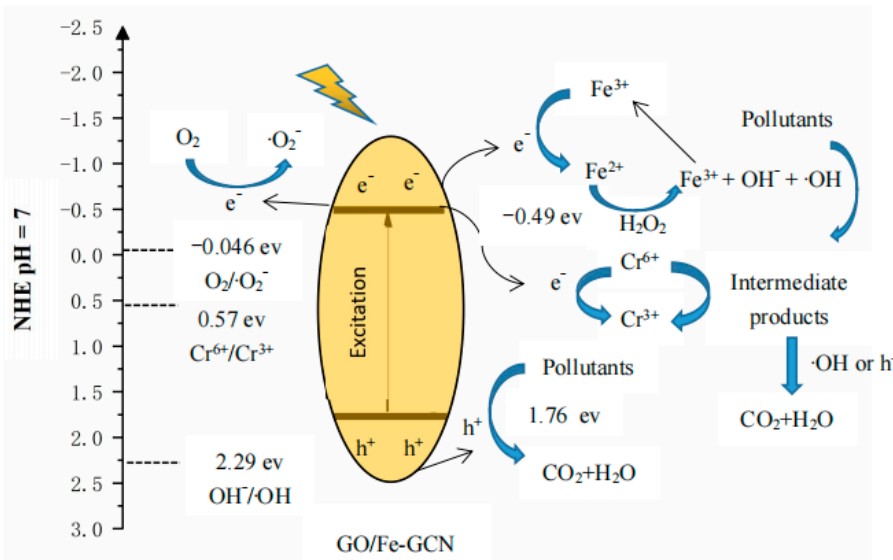

**Figure 10.** Proposed mechanism of the simultaneous Cr(VI) reduction and organic pollutants degradation in the photo-Fenton catalytic system.

The element iron is widely distributed on the earth, and iron composites are harmless to organisms and readily available. Meanwhile, iron species are also widely utilized in various fields [66,67]. As a typical organic semi-conductor material, g-$C_3N_4$ has several crucial advantages, such as being rich in N element, visible-light response, hypotoxicity, and a special two-dimensional structure. Moreover, g-$C_3N_4$ is a practical material for immobilizing transition metals by metal-N ligands, which provides a new strategic approach to assembling metal highly-dispersed heterogeneous catalytic materials with regulable valence [68]. Furthermore, graphene oxide (GO) can readily immobilize the surface and expose more active sites, which improves the electron transfer rate and further enhances the catalytic efficiency [69]. The above results demonstrate that blending the iron, g-$C_3N_4$, and GO into a ternary composite increased the active surface sites and adsorption capacity, leading to effective electron transfer and circulation of Fe(III)/Fe(II), which produced abundant hydroxyl radicals for the degradation of hazardous organic compounds.

## 3. Experimental Section

### 3.1. Synthesis

3.1.1. Synthesis of Graphene Oxide (GO)

The GO was synthesized using the Hummers method [47,48]. The whole procedure was carried out in three steps: (1) 2 g graphite powder was added into 46 mL concentrated $H_2SO_4$ (98%) and stirred for 24 h in a three-neck flask. (2) Then, the flask was cooled in ice water and meanwhile 12 g $KMnO_4$ was added to the flask. The reaction temperature did not exceed 20 °C. The flask with black product was then put into an oil bath, stirred at 40 °C for 30 min, and stirred at 90 °C for another 45 min. Distilled water (46 mL) was added to the flask for dilution, and the temperature was increased to 105 °C for 25 min with stirring. After cooling down to room temperature, 140 mL distilled water and 10 mL $H_2O_2$ solution (30%) were added, and a dark yellow solution appeared at this time. (3) The final process was purification: the powder was washed with 1 mol/L HCl solution to remove metal ions, followed by washing with water and anhydrous ethanol twice each. Finally, the powder was put into a dialysis bag for purification for a week with an appropriate amount of water, replacing the deionized water twice a day.

3.1.2. Synthesis of Graphite Phase Carbon Nitride (GCN)

Dicyandiamide (5 g) was put into an alumina crucible and calcined at 550 °C for 4 h under $N_2$ at a heating rate of 5 °C/min at 550 °C, to obtain massive GCN.

### 3.1.3. Synthesis of Iron-Doped Graphite Phase Carbon Nitride (Fe-GCN)

Fe(NO)$_3$·9H$_2$O with 5 g diccyanide and mass ratios of 0.15 were dissolved in a water bath at 80 °C (as shown in Figure 1a). The mixture was dried overnight at 80 °C. The obtained brown powder was put into an alumina crucible and calcined at 550 °C for 4 h under the protection of N$_2$ with an increase in temperature of 5 °C/min from room temperature. The product was denoted as Fe-GCN (As shown in Figure 1b). Furthermore, iron-doped GCN samples with different content of Fe were also synthesized for comparative experiments, the synthetic process is shown in Supplementary Notes S1.

### 3.1.4. Synthesis of Graphene/Iron-Doped Graphite Phase Carbon Nitride Composite (GO/Fe-GCN)

GO (0.05 g) was dispersed into 50 mL water and ultrasonication was used to evenly disperse it. Then, 0.25 g Fe-GCN was added and stirred for 30 min, and ultrasonic was performed for 1 h. This process was repeated for 4 times (as shown in Figure 1b).

### 3.2. Characterizations

A high-resolution transmission microscope (TEM) was employed to observe the particle morphology and element distribution of the samples on a ZEISS Gemini 300 (Tokyo, Japan), with working voltage of 200 kV, acceleration voltage of 0.02–30 kV, and probe beam of 3PA-20 NA. Instantaneous free radical determination was performed on a paramagnetic resonance instrument, Bruker EMX PLUS (Germany), with center field of 3502.00 G, sweep field width of 100.0 G, energy of 6.325 mW, and modulation frequency of 100.00 kHz. The qualitative phase and crystalline of samples were analyzed using an X-ray powder diffractometer (XRD Model: D/Max 2500PC) with Cu Kα target radiation (tube current I = 100 mA, tube voltage V = 40 kV), scanning angle range of 5~80°, scanning operation rate of 0.02°/s, and continuous scanning speed of 5°/min. A field emission scanning electron microscope (SEM, Zeiss SUPRA-55) was employed to observe the morphology, geometrical size, and dispersion of materials. The specific surface area, pore size, and pore volumes of samples were tested using a physical nitrogen adsorption–desorption instrument (ASAP 2460), and the N$_2$ adsorption–desorption isotherms of samples were obtained under −196 °C after vacuum degassing at 150 °C for 8 h. The pore size distributions of the samples were obtained using adsorption branch of adsorption–desorption isotherms. A UV-visible spectrophotometer (UV-2450) was utilized for measuring the light absorption range of the sample and calculating the band gap with a wavelength range of 200–1500 nm. Fluorescence photoluminescence spectrometer (PL) characterization (Perk in Elmer, Norwalk, USA) was used to measure the photoluminescence and electron hole separation ability of the samples (wavelength: 200–800 nm; excitation wavelength: 315 nm). X-ray photoelectron spectroscopy (XPS, Thermo Scientific K-Alpha, Waltham, USA) was used to analyze the surface elemental composition of the samples.

### 3.3. Adsorption and Photocatalytic Applications for Degradation

The degradation kinetics in the photocatalytic or photo-activated Fenton systems were quantitatively studied by fitting, and then fitting the fitted and kinetic models:

$$-\ln C_t/C_0 = kt \tag{5}$$

where C$_0$ (mg/L) is the initial concentration after dark adsorption is ignored, C$_t$ is the concentration at time T, and K (min$^{-1}$) is the pseudo first-order rate constant.

For exploring the effect of GO/Fe-GCN on the contaminant degradation performance under photocatalysis, GO/Fe-GCN (20 mg) and 100 mL 10 mg/L Rh B solution were added to 500 mL quartz reaction bottle under normal temperature and pressure. After dark adsorption for 30 min, a xenon lamp and magnetic stirrer were turned on. A 4 mL substitute test solution was taken at intervals of 30 min. The absorbance at 552 nm was measured. To explore the influence of GO/Fe-GCN on the contaminant degradation with the catalytic activity of photo-activated Fenton, 20 mg of GO/Fe-GCN was added into a

500 mL quartz reaction bottle under normal temperature and pressure. Then, 100 mL 10 mg/L contaminants solution was added. A certain amount of 30% $H_2O_2$ solution was also added after dark adsorption for 30 min. A xenon lamp was turned on, to stimulate the Fenton reaction. A 4 mL test solution was taken at intervals and separated from the granular catalyst by centrifuging, and the supernatant was extracted and measured at 552 nm by spectrophotometry. The concentration of Cu-EDTA was measured by spectrophotometric method at 730 nm, and the concentrations of total copper ions were measured using 700 series inductively coupled plasma-optical emission spectrometry (ICP-OES) from Agilent Technology. The $Cr^{6+}$ was measured through spectrophotometric determination of dibenzoyl dihydrazine at 540 nm.

## 4. Conclusions

The iron-doped g-$C_3N_4$/GO hybrid composite (GO/Fe-GCN) was successfully assembled and utilized as a high-efficiency adsorption–photo-coordinated Fenton heterogeneous catalyst. The GO/Fe-GCN possessed highly dispersed iron species, with a $Fe^{2+}$/$Fe^{3+}$ ratio value of 1.67. The well-established mesopores of 3.7 nm with a relatively higher specific area and pore volume benefited the reaction efficiency and contact of organic pollutants with active sites on the ternary catalyst of GO/Fe-GCN. As a typical n-type semiconductor, GO/Fe-GCN displayed a fast electron transfer rate, which was the reason for the high efficiency of the photo-generated carrier separation. In the static system, the photo-coordinated Fenton degradation rate of GO/Fe-GCN was enhanced to 98.71% within 20 min, because the excellent conductivity of GO accelerated the electron transport rate and $Fe^{2+}$/$Fe^{3+}$ circulation efficiency. In the dynamic adsorption-photo-coordinated Fenton system, the maximum removal rate of GO/Fe-GCN reached 96.5% and equilibrium at 83.6%. The GO component not only enhanced the adsorption, but also provided higher efficiency of photo-generated carrier separation and transport. The hybrid structure of GO/Fe-GCN and the high efficiency circulation of Fe(III)/Fe(II) played an essential role in the synergy of adsorption-enrichment and photo-coordinated Fenton reaction. GO/Fe-GCN also possessed a good capacity to treat complex water waste containing metallic ions, metal complexes, and organic pollutants, representing a promising water treating agent for application in pollution control.

**Supplementary Materials:** The following supporting information can be downloaded at: https://www.mdpi.com/article/10.3390/catal13010088/s1, Figure S1: SEM images for GO (a) and Fe-GCN (b); Figure S2: XRD patterns for GCN and Fe-GCN samples with different iron content; Figure S3: UV-vis DRS (a) and band-gaps (b) for GCN and Fe-GCN samples; Figure S4: Mott-Schottky curves for GO/Fe-GCN(a) and Fe-GCN(b), EST curve for GCN, Fe-GCN and GO/Fe-GCN(c), and photoluminescence intensity of GO/Fe-GCN and Fe-GCN (d); Figure S5: The visible light photocatalytic degradation rate of GCN towards Rh B of samples (a) and the quasi-first-order reaction kinetic rate of samples (b); Figure S6: The photo-coordinated Fenton degradation efficiencies of samples (a) and the quasi-first-order reaction kinetic rate of samples (b); Figure S7: The recycles of GO/Fe-GCN through photo-coordinated Fenton degradation; Figure S8: The photo-coordinated Fenton degradation efficiencies of GO/Fe-GCN towards single $Cr^{6+}$; Table S1: Band gap and energy band potentials of Fe-GCN and GO/Fe-GCN; Table S2: The specific surface areas ($m^2$/g), pore volumes($cm^3$/g) and BJH pore size (nm).

**Author Contributions:** Conceptualization, C.Y.; Methodology, A.C.; Formal analysis, Y.S.; Investigation, Q.Z., F.C. and H.G.; Data curation, L.W.; Writing—review and editing, H.M. All authors have read and agreed to the published version of the manuscript.

**Funding:** This study was funded by China Petroleum & Chemical Corporation (Sinopec) research project (321042 and 320005) and Science and Technology Project of Changzhou City (CE20215031).

**Data Availability Statement:** Not applicable.

**Acknowledgments:** We really appreciate graduated Qing Zhang's idea for the research, which inspired a new field.

**Conflicts of Interest:** The authors declare no conflict of interest.

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
