# Peer review of "Synergetic Adsorption–Photocatalytic Activated Fenton System via Iron-Doped g-C3N4/GO Hybrid for Complex Wastewater"

_catalysts, doi:10.3390/catal13010088_

Round 1
Reviewer 1 Report
In the research article entitled “Synergetic adsorption-photocatalytic activated Fenton system via iron-doped g-C3N4/GO hybrid with high efficient performance”, authors have investigated the photocatalytic ability of iron-doped g-C3N4/GO hybrid using Rh B organic dye. This work is interesting, but the article has many grammatical and sentence errors. The language organization needs to be improved. In addition, the authors should do the following before acceptance.
1. In the title, the authors have mentioned ‘high efficient performance’. Authors need to clarify what. rephrase the title.
2. Authors need to avoid terms like a novel, first report, etc. in the manuscript.
3. In the abstract, the authors have quoted as “the maximum removal rate of GO/Fe- 15 GCN reaches 96.5 % and equilibrium at 83.6%”. Against which pollutant? Mention.
4. Authors have also quoted that “GCN can also treat complex wastewater containing metallic ions, metal complexes and organic pollutants, which can facilitate the potential application in the treatment of water pollution”. But authors have studied only using dyes. The authors need to change the quote.
5. A detailed protocol for the synthesis of graphite phase carbon nitride (GCN) needs to be provided.
6. Authors need to improve the references by citing recent references.
https://doi.org/10.1007/s11356-022-23248-6
https://doi.org/10.1016/j.envres.2022.114270
https://doi.org/10.1016/j.seppur.2021.120313
https://doi.org/10.3390/catal12111489
https://doi.org/10.1007/s43630-022-00224-0
7. Results sections need to be subdivided to improve readability.
8. Reusability study for the photocatalyst to understand the economic feasibility of the applications.
9. Typographical errors can be avoided. The language and grammar used throughout the manuscript need to be improved
Reviewer 2 Report
Journal: Catalysts
Ms. ID.: catalysts-2112527
Title: Synergetic adsorption-photocatalytic activated Fenton system via iron-doped g-C3N4/GO hybrid with high efficient performance
Mao et al. aimed to synthesize and explore the mechanism of iron-doped g-C3N4/GO hybrid composite for a synergetic adsorption-photocatalytic activated Fenton system with high efficient performance. Combining iron-doped g-C3N4 with GO resulted in an efficient photo-activated Fenton catalyst that exerts the effects in a wide pH range. Fe2+ and Fe3+ were trapped by N-rich g-C3N4, which forms highly dispersed active sites. GO provides high adsorption capacity for g-C3N4/GO hybrid composite and promotes electron transfer via strong electronic coupling by sp2 bonding structure. It is a very interesting manuscript. It fits well with the scope of the Journal. The introduction is informative. The experimental section is detailed enough. The results are well presented. I consider the manuscript suitable for publication, but I also believe some improvements are needed. The list of specific issues that should be addressed is listed below.
-The title of Figure 3 is not visible.
-The discussion is good in general, but some additional comparison with other methods for Rh B degradation available in the literature would improve it.
Author Response
Mao et al. aimed to synthesize and explore the mechanism of iron-doped g-C3N4/GO hybrid composite for a synergetic adsorption-photocatalytic activated Fenton system with high efficient performance. Combining iron-doped g-C3N4 with GO resulted in an efficient photo-activated Fenton catalyst that exerts the effects in a wide pH range. Fe2+ and Fe3+ were trapped by N-rich g-C3N4, which forms highly dispersed active sites. GO provides high adsorption capacity for g-C3N4/GO hybrid composite and promotes electron transfer via strong electronic coupling by sp2 bonding structure. It is a very interesting manuscript. It fits well with the scope of the Journal. The introduction is informative. The experimental section is detailed enough. The results are well presented. I consider the manuscript suitable for publication, but I also believe some improvements are needed. The list of specific issues that should be addressed is listed below.
Ans: Thanks a lot for this so nice comments.
-The title of Figure 3 is not visible.
Ans: Thanks so much for the reminding.
The title of Figure 3 is Figure.3. XPS patterns for Fe-GCN (a) and GO/Fe-GCN (b): high-resolution C1s, N1s, O1s and Fe2p spectra.
-The discussion is good in general, but some additional comparison with other methods for Rh B degradation available in the literature would improve it.
Ans: This suggestion is very important and we really appreciate the preciseness of the reviewer. Our purpose is not only degradation the Rh B but also the complex wastewater contained organic pollutants, metallic ions and metal-complex. Besides that complex content, the degradation efficiency of Rh B is much higher than most of the treatment in ref.
Reviewer 3 Report
The authors fabricated a Fe doped C3N4/GO hybrid composite with enhanced photo-adsorption properties. The results are interesting. I am, however, not sure if the composite is really attractive and if the authors elucidated the fundamentals behind the efficient performance clearly.
Comment 1: Why this research is important to the field? Could authors provide more details? Appropriate and sufficient references should be cited in the introduction and discussion parts.
Comment 2: The writing needs to improve.
Comment 3: The GO/Fe-GCN showed a mesoporous structure, which, as elucidated by the author, was resulted from the oxidation process, could the author provide more information about it?
Comment 4: The photo adsorption data looks good, what about the repeatability of the experiment?
Comment 5: What is the specific role of each component, i.e. Fe, C3N4 and GO in the composite and how them worked synergistically?
Round 2
Reviewer 1 Report
All the queries raised have been addressed. Recommended for publication